The antibacterial mechanism of phenylacetic acid isolated from Bacillus megaterium L2 against Agrobacterium tumefaciens

Pan Hang 1
Xiao Yang 2
Xie Ailin 1
Li Zhu zhuliluck@163.com 1 3
Ding Haixia hxding@gzu.edu.cn 4
Yuan XiaoJu 5
Sun Ran 1
Peng Qiuju 1
1 Key Laboratory of Plant Resource Conservation and Germplasm Innovation in Mountainous Region (Ministry of Education), College of Life Sciences/Institute of Agro-bioengineering, Guizhou University , Guiyang , Guizhou Province , China
2 Institution of Supervision and Inspection Product Quality of Guizhou Province , Guiyang , China
3 Guizhou Key Laboratory of Agricultural Biotechnology, Guizhou Academy of Agricultural Sciences , Guiyang , China
4 Department of Plant Pathology, College of Agriculture, Guizhou University , Guiyang , China
5 Development Center of Planting , Huishui County of Qiannan Prefecture , Guizhou Province , China
Sobeh Mansour
Electronic publication date: 2022 Nov 8
Publication date: 2022
Volume: 10
Electronic Location ID: e14304
Received 2022 May 12; Accepted 2022 Oct 5
Copyright: ©2022 Pan et al.
Copyright year: 2022
Copyright holder: Pan et al.
License: This is an open access article distributed under the terms of the Creative Commons Attribution License, which permits unrestricted use, distribution, reproduction and adaptation in any medium and for any purpose provided that it is properly attributed. For attribution, the original author(s), title, publication source (PeerJ) and either DOI or URL of the article must be cited.
License URL: https://creativecommons.org/licenses/by/4.0/

Keywords: Bacillus megaterium, Metabolite, Phenylacetic acid, Agrobacterium tumefaciens, Antibacterial mechanism

Funding: Guizhou Province High-level Innovative Talent Project GCC[2022]027-1 The Science and Technology Project of Guizhou Province [2021]193 The National Natural Science Foundation of China 31660533 31460486 The Science and Technology Project of Guizhou Province [2017]5613 The Modern Industrial Technology System for Chinese medicinal materials in Guizhou Province This work was supported by the Guizhou Province High-level Innovative Talent Project (Qiankehe Platform Talent-GCC[2022]027-1), the Science and Technology Project of Guizhou Province (grant number [2021]193); the National Natural Science Foundation of China (grant numbers 31660533, 31460486), the Science and Technology Project of Guizhou Province (grant number [2017]5613); and the Modern Industrial Technology System for Chinese medicinal materials in Guizhou Province. The funders had no role in study design, data collection and analysis, decision to publish, or preparation of the manuscript.

==============================
Background

Agrobacterium tumefaciens T-37 can infect grapes and other fruit trees and cause root cancer. Given the pollution and damage of chemical agents to the environment, the use of biological control has become an important area of focus. Bacillus megaterium L2 is a beneficial biocontrol strain isolated and identified in the laboratory, which has a good antibacterial effect on a variety of plant pathogens. The antibacterial metabolites of L2 were separated and purified to obtain a bioactive compound phenylacetic acid (PAA).

Methods

The potential antibacterial mechanism of PAA against A. tumefaciens T-37 strain was determined by relative conductivity, leakage of nucleic acids, proteins, and soluble total sugars, sodium dodecyl sulfate-polyacrylamide gel electrophoresis (SDS-PAGE), and reactive oxygen species (ROS).

Results

PAA showed good antibacterial activity against strain A. tumefaciens T-37 with IC50 of 0.8038 mg/mL. Our data suggested that after treatment with PAA, the relative conductivity, nucleic acid, protein, and total soluble sugar of T-37 were increased significantly compared with the chloramphenicol treatment group and the negative treatment group. The total protein synthesis of T-37 cells was inhibited, the consumption of phosphorus decreased with the increase of incubation time, and the content of ROS was significantly higher than that in the negative treatment group. Meanwhile, the activity of two key enzymes (MDH and SDH) involved in the tricarboxylic acid cycle (TCA cycle) decreased. In addition, T-37 cells were found to be damaged by scanning electron microscopy observation. Our results showed that PAA can destroy cell membrane integrity, damage cell structures, affect cell metabolism, and inhibit protein synthesis to exert an antibacterial effect.

Conclusions

We concluded that the mechanism of action of the PAA against strain T-37 might be described as PAA exerting antibacterial activity by affecting cell metabolism, inhibiting protein synthesis, and destroying cell membrane integrity and cell ultrastructure. Therefore, PAA has a promising application prospect in the prevention and treatment of root cancer disease caused by A. tumefaciens.

Introduction

Plant diseases caused by plant pathogenic bacteria have a serious impact on the sustainable development of agriculture and lead to huge economic losses (Savary et al., 2012; Guo et al., 2021). Root cancer disease, also known as crown gall disease, is a global bacterial disease caused by the Gram-negative bacterium Agrobacterium tumefaciens which has a wide host range and can infect hundreds of different families of herbaceous and woody dicots plants, including shrubs, fruit, nut trees, ornamental trees, vine crops (grapevine), etc. (Otten, Burr & Szegedi, 2008; Guo et al., 2021). In addition, it can cause crown gall in many agriculturally and economically important species, such as Rosaceae (pear, cherry, apple, and rose), Vitaceae (grape) and Juglans (walnut) (Nonaka et al., 2019). Plants infected with the disease have a weak yellow shape, tumors, and a decreased fruit yield and ornamental value; the disease may even lead to plant death (Fuller et al., 2017). It is reported that crown gall disease is prevalent abroad, and it is also widely distributed in some northern provinces of China, such as Hebei, Henan, Shandong, Jilin, etc., causing grapevine, blueberries, Tectona grandis (teak), and cherries to be susceptible, with the highest incidence in nurseries reached 92%, causing severe economic losses (Alippi, Lopez & Balatti, 2010; Kuzmanović et al., 2018; Borges et al., 2019; Yan et al., 2020).

Furthermore, the mechanism of plant disease induced by A. tumefaciens is a topic of interest. A. tumefaciens has been reported to have the ability to integrate into the plant host genome by transfer of its T-DNA (White & Winans, 2007; Gohlke & Deeken, 2014; Bourras, Rouxel & Meyer, 2015; Guo et al., 2019). After transformation, the plant host cells direct the expression of T-DNA through a series of coordinated interactions to control the transitional synthesis of auxin and cytokinin, leading to the proliferation of plant cells and thus the growth of tumors called crown galls (White & Winans, 2007; Platt et al., 2014). In addition, infected plant tissue produces nutrients called opines to facilitate the growth of A. tumefaciens (Brencic & Winans, 2005; White & Winans, 2007). In general, the strategies of A. tumefaciens to infect plant hosts include (i) Agrobacterium attachment to the host cell, (ii) T-DNA processing and import into the host cell via a type IV secretion system, (iii) Similar to viral transport in mammalian cells, Agrobacterium T-DNA mediates its transport from the cytoplasmic to the nucleus using dynein motor proteins and host microtubule network proteins, (iv) Agrobacterium T-DNA targets host chromosomes through host transcription machinery, (v) T-strand is converted into double-stranded form by host DNA repair mechanisms and integrated into the host genome for expression, resulting in oncogenic reprogramming of cells, resulting in tumors (Citovsky et al., 2007; Bourras, Rouxel & Meyer, 2015).

At present, chemical agents such as fungicides and insecticides are mainly used for the prevention and control of the disease in production. However, they are prone to negatively impact the environment and even harm human health, therefore, the use of microbe-based biological agents has attracted significant attention (Sharma et al., 2020; Gomes et al., 2020). It has been reported that there are a variety of beneficial microorganisms that can be used to prevent and control such diseases, such as Rahnella aquatilis H2X, Agrobacterium vitis VAR03-1, and Bacillus spp., among which Bacillus spp. is the most important one type (Chen et al., 2007; Kawaguchi, Inoue & Ichinose, 2008; Yan et al., 2020). Bacillus spp. has a good antibacterial effect against pathogenic bacteria and can produce a variety of antibacterial metabolites, including surfactin, fengycin, iturin, bacillomycin, mycosubtilin, pumilacidin, etc. (Fira et al., 2018). Among them, surfactin, iturin A, and fengycin have good control effects on important agricultural plant diseases such as bananas and potatoes (Villegas-Escobar et al., 2013; Lin et al., 2018). Bacillus megaterium L2 with a good biocontrol effect was isolated from the Asilixi Steppe in Bijie City in the early stage of the laboratory, its antimicrobial activity against numerous plant pathogens such as Ralstonia solanacearum and Alternaria alternata reached more than 90% (Yang et al., 2016; Ji et al., 2019). Five monomer compounds were obtained from its fermentation crude extract by the bioactivity-guided separation and identified as phenylacetic acid, palmitic acid, erucamide, behenic acid, and β-sitosterol. These compounds have different degrees of antibacterial activity against Erwinia carotovora EC-1, A. tumefaciens T-37, and R. solanacearum RS-2, among which PAA showed the highest antibacterial activity against T-37, EC-1, and RS-2, all of which reached more than 80% (Xie et al., 2021).

Phenylacetic acid (PAA) is a volatile aromatic compound and plant growth hormone. Its molecular formula is C8H8O2, and the structure is shown in Fig. 1. PAA has good antibacterial activity and is widely present in organisms (Cook, 2019; Wu, Huang & Deng, 2020). As a natural auxin for plants, it can promote the growth of capsicum annuum, pea, and tobacco (Husain, Jain & Kothari, 1999; Cook, 2019). PAA can also be used as a potent bactericidal substance, which has inhibitory effects on both Gram-positive and Gram-negative bacteria and fungi, and exhibits strong antibacterial activity against a variety of plant pathogens (Kim et al., 2004; Somers et al., 2005; Wu, Huang & Deng, 2020). In addition, PAA is often used as a side-chain precursor for penicillinG production in a pharmaceutical factory (Harris et al., 2009). Besides plants, microorganisms including Bacillus, Enterobacter cloacae, Azospirillum brasilense, Rhizoctonia solani, and Burkholderia can also produce PAA (Kim et al., 2004; Slininger, Burkhead & Schisler, 2004; Somers et al., 2005; Bartz et al., 2013; Sopheareth et al., 2013; Akram, Anjum & Ali, 2016). Transamination of phenylalanine, decarboxylation of phenylpyruvate, and subsequent oxidation of phenylacetaldehyde have been reported as the most likely pathway for PAA biosynthesis in bacteria and fungi (Somers et al., 2005; Zhang et al., 2017). However, its specific regulatory mechanism is still unclear, and more research is needed to elucidate it.

Figure 1 The structure of PAA.

The previous laboratory work mainly focused on the isolation, purification, and antibacterial activity of bioactive compounds, and did not further explore the antibacterial mechanism of active compounds against sensitive strains. Moreover, to the best of our knowledge, few reports could be found on the possible antimicrobial mechanisms of PAA against some specific bacterium. Therefore, the objective of the current work was to specify the potential antimicrobial action of PAA against A. tumefaciens from three aspects: cell metabolism, biofilm integrity, and cell ultrastructure, and to further understand the antimicrobial mechanism of Bacillus.

Materials and Methods

Strains and culture medium

B. megaterium L2: provided by the Institute of Fungal Resources of Guizhou University, currently stored in the China Center for Type Culture Collection (CCTCC, NO. M2012381).

A. tumefaciens T-37: purchased from the Institute of Soil and Fertilizer, Chinese Academy of Agricultural Sciences, and preserved by the Institute of Fungal Resources Laboratory of the College of Life Sciences, Guizhou University.

Beef extract peptone medium (NA): peptone 10 g/L, NaCl 5 g/L, beef extract 3 g/L, agar powder 20g/L, distilled water 1000 mL, pH natural. All media were autoclaved at 121 °C for 15 min.

Experimental reagents and instruments

The experimental reagents and instruments used in this study are shown in Tables 1 and 2.

Table 1 The reagents used in this study.

Reagent	Purity	Manufacturer	
95% Ethanol	AR	Sangon Biotech (Shanghai) Co., Ltd.	
Concentrated sulfuric acid	AR	Dongguan Yuantao Chemical Co., Ltd.	
Glucose	AR	Tianjin Yongda Chemical Reagent Co., Ltd.	
Beef extract	AR	Beijing Aoboxing Biotech Co., Ltd.	
Peptone	BR	Beijing Aoboxing Biotech Co., Ltd.	
Agar powder	BR	Beijing Solarbio Science & Technology Co., Ltd.	
Nacl	AR	Chongqing Chuandong Chemical Group Co., Ltd.	
Coomassie Brilliant Blue R250	BS	Shanghai Jinsui Biotechnology Co., Ltd.	
Chloramphenicol	AR	Beijing Solarbio Science & Technology Co., Ltd.	
Ammonium persulphate	AR	Sinopharm Chemical Reagent Co., Ltd	
TEMED	BR	Beyotime Biotechnology Co., Ltd	
Bromophenol blue	AR	Tianjin Kermel Chemical Reagent Co., Ltd.	
Ammonium molybdate	AR	Wuhan Hezhong Biochemical Manufacturing Co., Ltd.	
Anthrone	AR	Tianjin Kermel Chemical Reagent Co., Ltd.	
NAD-Malate Dehydrogenase (NAD-MDH) Assay Kit	–	Beijing Solarbio Science & Technology Co., Ltd.	
Succinate Dehydrogenase (SDH) Assay Kit	–	Beijing Solarbio Science & Technology Co., Ltd.	
Reactive Oxygen Species Assay Kit	–	Beyotime Biotechnology Co., Ltd.	

Table 2 The instruments used in this study.

Instruments	Model	Manufacturer	
Conductivity Meter	DDB303A	Shanghai Precision Scientific Instrument Co., Ltd.	
Spectrometer	SpectroAmax TM 250	Molecular Devices.	
UV Spectrophotometer	BIOMATE 3S	Thermo Fisher Scientific.	
Scanning Electron Microscope	S-3400N	Hitachi, Ltd.	
Electrophoresis Apparatus	JY300HE	Beijing JUNYI Electrophoresis Co., Ltd.	
Electric Heating Incubator	HPX-9082MBE	Jiangsu Scientific Instruments and Materials Co., Ltd.	
Double-layer Extra Large Capacity Constant Temperature Culture Shaker	SKY-2112B	Shanghai DaPing Instrument Co., Ltd.	
Independent High Temperature Moist Heat Sterilizer	mLS-3750	Beijing Nuohuicheng Technology Co., Ltd.	

Isolation and purification of PAA

The steps of isolation and purification of PAA roughly include five steps. (i) The first is the large-scale fermentation of B. megaterium L2. (ii) The fermented bacterial cells were spray-dried to form a pulverized powder, and then extracted and concentrated by refluxing with ethanol at 80 °C to obtain the crude ethanol extract. (iii) Then, according to the antibacterial activity of the crude extract to the three indicator bacteria, fractional distillation was carried out with petroleum ether, ethyl acetate, n-butanol, and water to obtain the organic fraction with the highest activity, namely the ethyl acetate phase. (iv) The ethyl acetate extract was separated by silica gel column chromatography to obtain ten fractions of B1–B10. (v) Finally, the B4 fraction was separated by atmospheric pressure silica gel column chromatography (200-300 mesh) and recrystallization to obtain the monomeric compound PAA. For specific information, please refer to the literature (Xie et al., 2021).

Determination of IC50 of PAA

Preparation of bacterial suspension: T-37 strain was activated and transferred into the NA medium, and the cultures were incubated at 30 °C with shaking at 150 r/min for the logarithmic growth phase (18 h). Then, the T-37 strain culture was centrifuged at 6,000 r/min for 5 min to prepare bacterial suspension with sterile water, and the final cell concentration was adjusted to 1 × 108 CFU/mL approximately for further studies. PAA and chloramphenicol were dissolved in DMSO to make 10 mg/mL stock solutions. Then the suspension of the T-37 strain was inoculated into NA liquid medium at 10% inoculum volume, PAA was then added to NA liquid medium to final concentrations of 0.6, 0.8, 1.0, 1.2, and 1.4 mg/mL. The control was prepared as described above but in the absence of PAA. All samples were incubated at 30 °C, 150 r/min for 18 h, and the optical density (OD) value was measured at the maximum absorption wavelength of 400 nm. The sterile water was used as the negative control and chloramphenicol as the positive control, the inhibition rate was calculated according to the following formula, and the IC50 values of PAA and chloramphenicol were calculated by GraphPad Prism 9 software. The experiment was repeated 3 times. Inhibitionrate%=ODcontrol−ODtreatmentODcontrol×100

where OD control is the OD value of the negative control; and OD treatment is the OD value of PAA and chloramphenicol treatment, respectively.

Effects of PAA on cell membrane permeability of T-37 pathogenic bacteria

This analysis was performed using the method described by Song et al. (2016), with some modifications. T-37 bacterial suspension (1 × 108 CFU/mL) was inoculated into NA liquid medium at 10% inoculum volume, and the PAA was added to make the final concentration IC50. The bacteria cultures were incubated at 30 °C, with shaking at 150 r/min. Samples were taken out after being treated for 0, 1, 2, 3, 4, and 5 h, then 5 mL of the bacterial culture was centrifuged at 4,000 r/min for 10 min, and the supernatant was taken to measure the relative conductivity. Sterile water and chloramphenicol were used as the negative control and positive control, respectively. All tests were performed in triplicate.

Effects of PAA on cell membrane integrity of T-37 pathogenic bacteria

Nucleic acid and protein leakage

Nucleic acid and protein leakage analysis was performed following the method described by Zhou et al. (2020), with some modifications. The bacteria suspension of T-37 was inoculated into NA liquid medium at 10% inoculum volume, and PAA was added to make the final concentration IC50 value. The bacteria cultures were incubated at 30 °C, with shaking at 150 r/min. During the culture, six samplings were carried out at 0, 2, 4, 6, 8, and 10 h, and all samples were centrifuged at 4,000 r/min for 10 min and measured the absorbance at 260 nm and 280 nm in the supernatant. Sterile water and chloramphenicol were used as the negative control and positive control, respectively. All tests were performed in triplicate.

Leakage of total soluble sugar concentration

The determination of the total sugar of the sample was conducted according to the method of Zhou et al. (2019). We draw first the standard curve using the concentration of glucose as the abscissa (mg/mL) and the absorbance as the ordinate. Then after, the bacteria suspension of T-37 was inoculated into NA liquid medium at 10% inoculum volume, and the PAA was added to make the final concentration IC50. The bacteria cultures were incubated with shaking (150 r/min) at 30 °C. After 0, 2, 4, 6, 8, 10, and 12 h of incubation, aliquots of 1 mL were taken out and centrifuged, then took 50 µL of supernatant and added to 200 µL of anthrone reagent. The mixture was cooled on ice for 5 min and boiled for 10 min immediately. After it was naturally cooled to room temperature, the absorbance was measured with SpectroAmax TM 250 at 630 nm using sterile water as the negative control and chloramphenicol as the positive control. All tests were performed in triplicate.

SDS-PAGE analysis of total protein of T-37 bacteria

This analysis was performed using the method of Zhao et al. (2015), with some modifications. The bacterial suspension containing IC50 concentration PAA was prepared according to the previous method. The bacteria cultures were incubated at 30  °C, with shaking at 150 r/min. After 0, 4, 8, and 12 h of incubation, 1 mL of bacterial culture was taken out, centrifuged at 4,000 r/min for 10 min, and then the supernatant was discarded, and the precipitate was washed twice with sterile water. Then added 40 µL of 2 × loading buffer into the bacterial suspension, heated at 100 °C for 10 min, and centrifuged to collect the supernatant. Sterile water and chloramphenicol were used as the negative control and positive control, respectively. The SDS–PAGE was performed with a 12% separating gel, 5% stacking gel, and 5 × Tris-glycine electrode buffer, the sample volume was 10 µL, the voltage was 85 V, the current was 100 mA, and the time was 1.5–2 h. Then the gel was stained with 0.5% Coomassie brilliant blue R-250 was stained overnight, and then decolorized with a decoloring agent (ethanol-acetic acid-water solution, 10:5:85, v/v) to obtain the separated protein bands.

Effects of PAA on phosphorus metabolism in T-37 bacteria

The determination of the phosphorus content of the sample was conducted according to the method of Zhai et al. (2006). Bacterial cells were collected by centrifugation at 4000 r/min for 10 min, washed twice with sterile water, and diluted the bacterial cells to OD600 = 0.5. Added 0.5 mL bacterial culture and 0.5 mL glucose solution (1.0 mg/mL) to the test tube, mixed with 200 µL phosphoric acid standard solution (500 µg/mL), and then bacterial cells were mixed again with PAA to make the final concentration of IC50. When cultured for 0, 2, 4, 6, 8, 10, and 12 h, 0.1 mL of bacterial culture was added to 1 mL of trichloroacetic acid-ferrous sulfate solution, which was reacted at room temperature for 10 min, and centrifuged at 4,000 r/min for 10 min. The ammonium molybdate solution (50 µL) was added to 0.2 mL of supernatant, shaken well, incubated at 30 °C with a water bath for 15 min, and cooled naturally to room temperature, then measured the absorbance with SpectroAmax TM 250 at 630 nm. Sterile water and chloramphenicol were used as the negative control and positive control, respectively. All tests were performed in triplicate.

Effects of PAA on the content of reactive oxygen species (ROS) in T-37 cells

The bacteria suspension of T-37 was inoculated into NA liquid medium at 10% inoculum volume and cultured at 30 °C for 18 h. The content of ROS was determined according to the kit instructions. Cells from the bacteria culture were collected by centrifuged for 5 min at 12,000 r/min, washed twice with 0.1 M phosphate buffer solution (PBS, pH 7.4), and dissolved with 10 µmol/L 2, 7-dichlorofluorescein diacetate (DCFH-DA). All samples were incubated at 37 °C for 30 min, shaken for 3–5 min, and then washed the cell pellet with PBS three times to remove extracellular residual DCFH-DA. Finally, added PAA to the prepared solution to make the final concentration of IC50. Fluorescence intensity was detected by a fluorescence microscope every 10 min. The fluorescence results were quantitatively analyzed by Image J 17.0 software, and the fluorescence intensity was calculated according to the following formula. Fluorescenceintensity=FluorescenceintensitywithinthesamplingareaSamplingarea.

Effects of PAA on the activity of key enzymes of T-37 tricarboxylic acid

The bacteria suspension of the T-37 strain was inoculated into 100 mL of fresh culture medium at 10% inoculum volume in the presence of PAA at the IC50 concentration, and the negative control was prepared as described above, but in the absence of PAA, chloramphenicol was used as the positive control, all samples were incubated at 30 °C for 18 h. After treatment, the activities of succinate dehydrogenase (SDH) and NAD-malate dehydrogenase (NAD-MDH) were determined by the SDH assay kit and NAD-MDH assay kit (Solarbio, Beijing, China) respectively according to the manufacturer’s guidance. The experiments were performed in triplicate.

Scanning electron microscopy (SEM) analysis

This analysis was performed using the method of Zhang et al. (2016), with some modifications. The bacterial suspension containing IC50 concentration PAA was prepared according to the previous method. The control (without PAA) and samples were incubated at 30 °C, 150 rpm for 18 h, and centrifuged at 4,000 r/min for 10 min after cultivation, discarded the supernatant, and washed the precipitate twice with PBS for use. Scanning electron microscope (SEM) sample processing: the samples were fixed with 2.5% glutaraldehyde at 4 °C for 10 ∼12 h, washed twice with 0.1 M PBS, and then dehydrated in gradient ethanol solutions of 30%, 50%, 70%, 90%, and 100%. Then the 100% ethanol solution was replaced by tert-butanol and fixed for 10 min. Finally, all the samples were plated with a metal film in the vacuum evaporator and observed by SEM (Hitachi S-3400N, Hitachi, Japan).

Experimental data processing

Excel 2016 was used for data processing, and IBM SPSS Statistics 26 software was used for all statistical analyses. The data were analyzed by analysis of variance (ANOVA) and Duncan test, besides, the calculation of the IC50 value and graphs were produced using Graph Pad Prism 9.0, and Image J 17.0 software was used for the conversion of fluorescence intensity and gray value. P < 0.05 was considered statistically significant.

Results

IC50 of PAA determination results

In this experiment, the inhibition rate of PAA on T-37 was determined at five concentrations of 0.6 mg/mL, 0.8 mg/mL, 1 mg/mL, 1.2 mg/mL and 1.4 mg/mL. Obtained results (Fig. 2) showed that at the inhibition rate of PAA reached 80.00 ± 0.02% at the mass concentration of 1.0 mg/mL and no significant difference were observed at concentration of 1.2 mg/mL and 1.4 mg/mL (P > 0.05). While the inhibition rate of chloramphenicol reached 87.76 ± 0.01% at the concentration of 0.2 mg/mL and was not significantly different from that at 0.4 mg/mL (P > 0.05). Concerning the IC50 calculated by Graph Pad Prism 9 software was 0.8038 mg/mL and 0.0658 mg/mL respectively for the PAA and the chloramphenicol and those values were for subsequent experiments.

Figure 2 The IC50 of PAA against A. tumefaciens T-37.

Examination of relative electric conductivity to indicate changes in the permeability of T-37 bacterial cell membranes

The results in Fig. 3 showed the effect of compound PAA on the membrane permeability of T-37 cells by relative electric conductivity assay. Our data suggested that there was less change in the relative electric conductivity during the whole experiment period in the negative control and positive control groups, and the relative electric conductivity of T-37 cells treated with PAA showed an upward trend at 0∼3 h, but it began to decrease for 4 h. Specifically, after treatment for 1 h, there was a significant difference in the extracellular relative electric conductivity between the PAA group and the negative group (P < 0.05), indicating that PAA began to destroy the cell membrane at this stage, and the intracellular electrolyte was released, leading to an increase in the relative electric conductivity. At 3 h, the relative electric conductivity of the PAA treatment group increased by 8.87% and 10.55% compared with the chloramphenicol treatment and the negative treatment group, respectively (P < 0.05). The relative electric conductivity of the bacteria cells treated with PAA decreased slowly at 4∼5 h, but it was still higher than in other treatment groups. These results indicated that PAA could affect the permeability of the T-37 cell membrane and release the cell contents, which led to the increase of electric conductivity in the extracellular environment.

Figure 3 Relative electric conductivity changes of A. tumefaciens T-37 after treatment with PAA and Chloramphenicol.

CK represents the negative control. Data are expressed as the average of three replicate experiments. Error bars represent mean ± SD.

Effects of PAA on cell membrane integrity of T-37 pathogenic bacteria

Leakage of T-37 bacterial nucleic acid and protein

The integrity of the cell membrane was determined by measuring the release of protein (OD260) and nucleic acid (OD280) of the supernatant of tested bacteria. The nucleic acid leakage of T-37 cells caused by PAA was shown in Fig. 4A. After treatment with PAA, the absorbance value increased linearly. At 2 h, the nucleic acid leakage of T-37 cells was not significantly different from the negative control, which indicated that the cell membrane was not seriously damaged and there was no obvious leakage of nucleic acid. Compared with the negative control, the leakage of nucleic acid treated with PAA increased by 1.32 times for 4 h (P < 0.05). In addition, after 10 h of incubation, the nucleic acid leakage of the PAA treatment group increased by 25% and 70.45% compared with the negative control and the positive control group, respectively, indicating that the cell membrane integrity of the T-37 cells treated with PAA was destroyed, resulting in the leakage of intracellular nucleic acids. Moreover, the protein leakage of T-37 cells caused by PAA was shown in Fig. 4B. The PAA treatment group was significantly higher than the negative treatment group at 2 h (P < 0.05), and the absorbance value showed an upward trend with the extension of the culture time, indicating that the cell membrane of T-37 cells had been destroyed at 2 h. At 10 h, the OD280 of protein treated with PAA was 0.082 ± 0.005, which increased by 36.67% and 74.47% compared with the chloramphenicol treatment group and the negative treatment group (P < 0.05). The above results demonstrated that the integrity of the cell membrane was severely damaged at this time, resulting in a large amount of intracellular protein leakage.

Figure 4 Nucleic acid and protein leakage of A. tumefaciens T-37 after treatment with PAA and chloramphenicol.

Data are expressed as the average of three replicate experiments. Error bars represent mean ± SD. Different lowercase letters mean significant differences, and the lowercase letters used for the significance comparisons apply only to comparisons within the same time period (P < 0.05). (A) Nucleic acid leakage; (B) protein leakage.

Leakage of total soluble sugar in T-37 bacteria

The leakage of total soluble sugar in T-37 cells caused by PAA was shown in Fig. 5. The total sugar concentration in the culture medium of the blank treatment group showed a gradual downward trend with time, which can be described as the continuous consumption of sugar substances during cell growth. The leakage of total soluble sugar from T-37 cells treated with PAA and chloramphenicol increased as the treatment time up to 2 h with significant differences (P < 0.05), indicating that the T-37 cell membrane had been damaged at this time. And the concentration of total soluble sugar in the extracellular environment increased with the increase of treatment time and reached the maximum at 10 h (0.2430 ± 0.0047 mg/mL), which increased by 29.74% and 144.47% compared with the chloramphenicol group and the negative treatment group, respectively (P < 0.05). These findings indicated that the integrity of the cell membrane of pathogenic bacteria T-37 was severely damaged, and intracellular sugar substances leaked into the medium, increasing the concentration of total soluble sugars in the medium.

Figure 5 Total soluble sugar leakage of A. tumefaciens T-37 after treatment with PAA and chloramphenicol.

Data are expressed as the average of three replicate experiments. Error bars represent mean ± SD. Different lowercase letters mean significant differences, and the lowercase letters used for the significance comparisons apply only to comparisons within the same time period (P < 0.05).

SDS-PAGE total protein profile of T-37 bacteria

The SDS-PAGE spectrum of the soluble protein of T-37 cells treated with PAA is shown in Fig. 6. The protein profiles of T-37 cell treated with PAA differed from those of the control. After treatment with PAA at IC50 concentration for 4 h, the protein bands of the negative control group showed strong intensities, but the bacterial protein bands of the PAA treatment group got much fainter, and some even disappeared. Similar results were also found in bacteria cells treated with chloramphenicol. After treatment for 8 h, the protein bands in the PAA treatment group almost disappeared. The above results indicated that the reason for the disappearance of protein bands’ might be PAA interfered synthesis and expression of T-37 cell proteins, or resulted in the leakage of proteins from bacterial cells.

Figure 6 SDS-PAGE patterns of total proteins for A. tumefaciens T-37 treated with PAA and chloramphenicol.

M represents marker; C, PAA and Chlor represent the negative group, phenylacetic acid treatment group and chloramphenicol treatment group treated for 0 h,4 h, 8 h, 12 h, respectively (follow left to right).

Phosphorus metabolism in T-37 bacteria

The overall metabolic function and growth of cells can be reflected by measuring phosphorus consumption in microbial metabolic activities. It can be seen from Fig. 7 that the phosphorus concentration in the three treatment groups showed a downward trend with the incubation time, indicating that the pathogenic bacteria T-37 consumed phosphorus for normal metabolic activities during the cultivation process. However, in the PAA and chloramphenicol treatment groups, the phosphorus consumption of T-37 cells decreased with the increase in incubation time, which indicated that phosphorus metabolism was severely affected by PAA.

Figure 7 Phosphorus metabolism change of A. tumefaciens T-37 after treatment with PAA and chloramphenicol.

Data are expressed as the average of three replicate experiments. Error bars represent mean ± SD.

Reactive oxygen species (ROS) content of T-37 bacteria

The effect of PAA on the ROS content of pathogenic bacteria T-37 was shown in Fig. 8. PAA can stimulate T-37 cells to produce more ROS, and the ROS content level in the cytoplasm of the PAA treatment group showed a trend of first increasing and then decreasing, reaching a peak value at 30 min of treatment, which was 1.112 ± 0.064, an increase of 45.42% compared with the negative treatment group (P < 0.05). After being incubated for 40 min, chloramphenicol reached the peak value, which was 0.861 ± 0.016, an increase of 16.75% compared with the negative treatment group (P < 0.05). In summary, PAA can increase the ROS content in T-37 cells, which may damage the cellular structure of T-37 cells.

Figure 8 The ROS content of A. tumefaciens T-37 after treatment with PAA and chloramphenicol.

Data are expressed as the average of three replicate experiments. Error bars represent mean ± SD. Different lowercase letters mean significant differences, and the lowercase letters used for the significance comparisons apply only to comparisons within the same time period (P < 0.05). Rosup: reference.

MDH and SDH tricarboxylic acid key enzyme activities in T-37 bacteria

The antibacterial action of PAA may be partly due to its effect on key enzyme activities of T-37 cells. By measuring the two key enzyme activities of MDH and SDH in the tricarboxylic acid cycle (TCA cycle) of pathogenic bacteria T-37 treated with PAA, it can reflect the energy metabolism status. As we can see from Fig. 9A, the MDH enzyme activity of strain T-37 in the negative group was at a normal level. Compared with chloramphenicol and PAA treatment, the enzyme activity of MDH in T-37 cells was significantly different (P < 0.05). The changes in SDH enzyme activity in the negative control group and treatment groups (both PAA and chloramphenicol) exhibited the same trend (Fig. 9B). These results indicated that PAA can decrease effectively the enzymatic activities of MDH and SDH in the TCA cycle, resulting in the influence of the energy metabolism pathway of T-37 cells, thereby inhibiting its normal physiological metabolism.

Figure 9 Key enzyme activities of A. tumefaciens T-37 TCA cycle after treatment with PAA and chloramphenicol.

Data are expressed as the average of three replicate experiments. Error bars represent mean ± SD. Different lowercase letters mean significant differences (P < 0.05). (A) MDH enzyme activity; (B) SDH enzyme activity.

Scanning electron microscope (SEM) observation of T-37 cells

SEM micrographs revealed changes in morphology and the outer surface of bacterial cells. Figure 10 showed the SEM photomicrographs of T-37 cells treated with and without PAA. As observed in Fig. 10A, untreated cells showed typical bacilliform morphology with a plump and smooth surface, no swelling and shrinkage, and were uniform in distribution, indicating that the cell membrane was intact and the cells were in normal condition. In contrast, T-37 cells treated with PAA at IC50 concentration revealed a severely damaging effect on the cell morphology. It can be seen from Fig. 10B that adhesion occurs between the cells of strain T-37, the boundaries were blurred, the cells were deformed and tarnished, and the cells can be seen to be damaged and broken, some bacterial cells appeared dented, and the cells were obviously damaged, indicating that PAA could destroy the morphological structure of strain T-37 cells. These findings demonstrated that PAA treatment may result in damage to the T-37 strain cell wall and cytoplasmic membrane and further inhibit cell growth.

Figure 10 SEM of A. tumefaciens T-37 untreated cells and after treatment with PAA.

(A) T-37 negative treatment group; (B) PAA treatment group.

Discussion

As a phytopathogenic bacterium, A. tumefaciens is one of the main pathogens causing crown gall disease in plants, which brings great losses to the agricultural economy (Fuller et al., 2017). B. megaterium is one of the commonly used strains in biological control, which has broad spectrum-antimicrobial properties, and it can produce a wide range of secondary metabolites to inhibit the growth of pathogenic bacteria and fungi, which can be used to control numerous plant diseases (Mannaa & Kim, 2018; Acurio Vásconez et al., 2020; Wang et al., 2020; Saleh et al., 2021). PAA was isolated from the crude extract of B. megaterium L2 in our laboratory, and the IC50 was 0.8038 mg/mL. The extracellular relative electric conductivity of the T-37 cells treated with PAA increased by 10.55% compared with the negative control group, indicating that PAA could affect the permeability of the T-37 cell membrane and release the cell contents into the culture medium for conductivity rises. At 10 h, the nucleic acid and protein at 260 nm and 280 nm were 0.075 ± 0.002 and 0.082 ± 0.005, respectively, which increased by 70.45% and 74.47% compared with the negative treatment group, meanwhile, the total soluble sugar concentration reached its peak value at 10 h, which was 0.2430 ± 0.0047 mg/mL, an increase of 144.47% compared with the negative treatment group. The above results indicated that PAA could destroy the permeability and integrity of the T-37 cell membrane. The SDS-PAGE protein profiles showed that the protein bands of T-37 cells treated with PAA tended to disappear for 8 h, which significantly affected the protein synthesis of T-37 cells. While in terms of cell metabolism, when T-37 was treated with PAA, its phosphorus consumption decreased with the increase in incubation time, and the content of ROS reached a maximum of 1.112 ± 0.064 at 30 min, an increase of 45.42% compared with the negative treatment group. At the same time, the enzymatic activities of MDH and SDH in the TCA cycle decreased, indicating that PAA significantly affected the cellular metabolism of T-37 cells. In addition, the SEM observation of the ultrastructure of T-37 cells treated with PAA showed that the cells were adhered and damaged, indicating that PAA can destroy the morphological structure of T-37 cells.

In this study, the mechanism of antibacterial action of PAA was confirmed based on the results of relative electric conductivity, the release of cellular components, protein synthesis, phosphorus metabolism, intracellular ROS content, key enzymes of the TCA cycle, and SEM. The antibacterial mode of action of PAA was first revealed by measuring cell membrane permeability by relative electric conductivity. The cell membrane is a kind of semi-permeable membrane. As a protective barrier of cells, it can selectively control the exchange of substances between cells and the external environment and is the defense line for microorganisms to resist the damage caused by external substances. When bacterial cells are subjected to the action of antibacterial substances, the cell membrane is damaged and loses its function, causing the exudation of intracellular substances, especially small molecular substances, such as Na+ K+ and PO43-, the first leak out of the cell, leading to an increase in the relative conductivity. This means that changes in extracellular relative electric conductivity can reflect whether the permeability of the bacterial cell membrane is affected (Tang et al., 2009; Lv et al., 2011; Diao et al., 2014; Zhang, Zhang & Xu, 2020). In addition, bacterial intracellular nucleic acids, proteins, and soluble sugar will leak, destroying bacterial structures and accelerating their death (Kohanski, Dwyer & Collins, 2010). Our results (Fig. 3) showed that after PAA treatment of T-37 cells, the relative electric conductivity of the extracellular environment was significantly increased compared with the control group, which was consistent with the research results, which indicated that PAA affected the cell membrane permeability of T-37 (Zhang et al., 2016).

Significant increases in optical densities at 260 nm and 280 nm were observed with increasing incubation time (Figs. 4A and 4B), revealing the leakage of cellular components, a commonly used indicator to study the inhibition of pathogenic bacteria by antibacterial compounds, which can reveal the integrity of the cell membrane (Yi et al., 2014; Zhang et al., 2016; Mutlu-Ingok & Karbancioglu-Guler, 2017; Zhang, Zhang & Xu, 2020). On the other hand, the results of total soluble sugar showed that the content of total soluble sugar increased after PAA treatment of T-37 (Fig. 5), which was consistent with the results reported in the study, indicating that the integrity of the cell membrane was destroyed, and bacteria could not use sugar substances to die (Qian, Tao & Xie, 2010; Zhou et al., 2019; Zhang, Zhang & Xu, 2020). These results indicated that the destruction of the cell membrane integrity of T-37 cells resulted in the release of its nucleic acids, proteins, and sugar substances to the outside of the cell.

The mechanism of antibacterial action of PAA was also confirmed in the results of inhibiting protein synthesis. Protein is the basic organic matter that constitutes a cell, participates in almost all life metabolic activities in the cell, and is the material basis and main undertaker of life activities. SDS-PAGE gel electrophoresis can reflect the change of protein expression (Fig. 6), which was consistent with the results of bacterial cells treated with antibacterial compounds such as sugar fatty acid esters and glycine, indicating that PAA can inhibit the synthesis of T-37 bacterial proteins and expression to inhibit its growth and proliferation (Sitohy, Mahgoub & Osman, 2012; Zhao et al., 2015).

In addition to the mechanism of antibacterial action of PAA in cell membrane permeability and integrity, it has also been confirmed in cellular metabolism. Phosphorus is a trace element required by all organisms, an important component of nucleic acid, phospholipid, and sugar metabolism intermediates, and is indispensable in the energy metabolism of cells. Microorganisms can use glucose and through a series of phosphorylation to provide the required energy (Qian, Tao & Xie, 2010). According to Zhou et al. (2019), Quercus variabilis Blume extract not only disrupted the cell membranes of Salmonella paratyphi A and Staphylococcus aureus but also severely affected their phosphorus metabolism, our results (Fig. 7) were consistent with it. ROS is a chemical substance formed after incomplete reduction of oxygen with strong oxidative activity, mainly including superoxide anion radical, hydrogen peroxide, singlet oxygen and hydroxyl radical, etc. (Yang, Chen & Shi, 2019). In addition, ROS plays an important role in participating in energy metabolism, material exchange, and regulating various physiological functions of organisms. When ROS is excessively produced, it will cause oxidative stress in the organism, induce apoptosis, and affect the normal physiological activity of cells (Li et al., 2016; Qian et al., 2016; Villalpando-Rodriguez & Gibson, 2021). The level of ROS in the organism is in a state of equilibrium under normal circumstances, however, when the balance is disrupted, it will cause oxidative damage to a wide range of biomolecules including DNA. For instance, it has been reported that ROS can produce around 100 different types of oxidative damage bases that severely disrupt intracellular homeostasis (Cadet & Wagner, 2013; Metcalfe & Olsson, 2021). The results of ROS from the present study showed that PAA could stimulate the production of more ROS in T-37, reaching the peak at 30 min, an increase of 45.42% compared with the negative treatment control, which was consistent with the research report (Guo et al., 2021). These results suggest that PAA can induce oxidative stress in T-37 cells to accumulate a large amount of ROS, damage cell structure, and accelerate cell death.

Based on the antibacterial effect of PAA on T-37 intracellular ROS, the activities of SDH and MDH in the TCA cycle were determined to reveal the mechanism of antibacterial action of PAA on key enzymes in T-37 cell metabolism. TCA cycle is a ubiquitous metabolic pathway in aerobic organisms. It is the final metabolic pathway of carbohydrate, lipid, and amino acid, and also the hub of the metabolic connection of the three nutrients. MDH is a class of enzymes widely present in living organisms, mainly involved in the dehydrogenation reaction of malate to oxaloacetate (OAA). OAA formed by MDH is a key intermediate metabolite of many important metabolic pathways including the TCA cycle, glyoxalic bypass, amino acid synthesis, and gluconeogenesis, and is one of the key oxidoreductases in the central metabolic pathway of cells. In addition, it is also involved in the C4 cycle, gluconeogenesis, fatty acid oxidation, nitrogen assimilation, amino acid synthesis, and other metabolic activities (Lee, Hong & Kim, 2019). SDH is the key enzyme of the TCA cycle, also known as respiratory chain complex II or succinate - ubiquinone oxidoreductase, which can reduce ubiquinone to panthenol through electron transfer during the conversion of fumarinate and succinic acid, resulting in oxidative phosphorylation and energy release (Saraste, 1999). As expected, consistent with our results (Fig. 9), PAA can reduce the enzymatic activities of MDH and SDH in the TCA cycle, which affects the normal physiological metabolic activity of T-37 cells. Moreover, since SEM micrographs can intuitively reflect the morphological changes of the bacterial membranes, the use of electron microscopes to observe the morphological changes of pathogenic bacteria after drug treatment is a common method to study the antibacterial mechanism (Zhang et al., 2016). Cells are the basic structural and functional units of living organisms. Observing the cell surface, especially the damage to the cell membrane, is strong evidence to judge whether the drug has an inhibitory effect on the cell (Chen & Cooper, 2002). The SEM results (Fig. 10) showed that PAA can damage the morphological structure of strain T-37 cells, thereby inhibiting their normal growth, which was consistent with research reports (Zhang et al., 2016; Zhou et al., 2019; Zhang, Zhang & Xu, 2020).

Bacteria are among the important infection sources of plant diseases. Helping plants defend against bacterial infestation has become an imminent challenge due to the rapid spread of multi-drug resistant bacteria (Moghadamtousi et al., 2014; Vasconcelos, Croda & Simionatto, 2018). The principle of synergy among antimicrobial compounds has been reported to be that, the combined use of antimicrobial compounds may enhance efficacy, increase bioavailability, decrease adverse side effects, lower the required doses, reduce toxicity, and reduce the occurrence of antimicrobial resistance (Cottarel & Wierzbowski, 2007; Inui et al., 2007). New antipathogenic combination drugs which include natural product combinations have recently become a research focus (Van Vuuren & Viljoen, 2011). Studies on the antibacterial mechanisms of essential oils, cinnamon, and its constituents have shown that mixtures of different constituents can produce synergistic or additive effects, resulting in enhanced antibacterial effects (Bassolé & Juliani, 2012; Vasconcelos, Croda & Simionatto, 2018). For instance, essential oils with phenols or aldehydes such as citral, cinnamaldehyde, eugenol, carvacrol, or thymol as the main components have the highest antibacterial activity (Dorman & Deans, 2000; Ben Arfa et al., 2006; Pei et al., 2009; Bassolé et al., 2010; Bassolé & Juliani, 2012). In our laboratory, five monomer compounds with strong antibacterial activity, which include PAA, palmitic acid, erucamide, behenic acid, and β-sitosterol, were isolated from B. megaterium L2 (Xie et al., 2021). This promising result made it a good candidate to enhance the inhibitory effect of existing antimicrobial agents by synergistic action.

At present, there are more than 20 manufacturers producing PAA in China, with a production capacity of about 18,000 t/a and demand of about 11,000 t/a (Han et al., 2017). As an important organic chemical raw material, PAA is mainly used to improve the output of penicillinG (Harris et al., 2009). In other respects, PAA may not only be used in the pesticide industry to make insecticides, bactericides, herbicides, etc. but also can be used as a fixative or modifier in the fragrance industry, widely used in soap, detergent, cosmetics, tobacco, food, etc. (Liu & Cao, 2011; Han et al., 2017). In addition, PAA can also be used in the manufacture of high-performance engineering plastics, fluorescent brighteners, dyes, liquid crystal material, and organic synthesis (Liu & Cao, 2011; Han et al., 2017).

Presently, the production method of PAA at home and abroad is mainly the hydrolysis method of phenyl acetonitrile. However, the production process of this method will produce highly toxic substances to pollute the environment, so it becomes more and more important to adopt safe and environmentally friendly biosynthesis methods. Unfortunately, there are no reports of microorganisms producing PAA through natural fermentation have been retrieved so far. In the previous work of the laboratory, the production of PAA was further increased by ARTP mutagenesis of B. megaterium L2 (unpublished), which revealed the possibility of producing PAA by microorganisms in the future. In addition, it has been reported that PAA can be produced by genetically or metabolically engineered strains, please refer to the literature for more details (Koma et al., 2012; Zhang et al., 2017). However, the aldehyde dehydrogenase pathway involved in the final step to generate PAA has not been experimentally characterized, and this gap limits the process development for producing PAA from phenylalanine, suggesting a direction for future work.

Conclusions

In conclusion, this study combined relative electric conductivity, cell component leakage, SDS-PAGE, intracellular ROS, the key enzyme activity of TCA and SEM observation showed that PAA was an effective antibacterial compound of the T-37 strain. The results from the present study also demonstrated the antibacterial mechanism of PAA against T-37 strain by affecting cell metabolism, cell membrane integrity, inhibiting protein synthesis, and destroying the cell ultrastructure to further inhibit cell growth or lead to cell death. Therefore, the PAA produced by B. megaterium L2 has a good application prospect for the prevention and treatment of root cancer disease.

Supplemental Information

Data S1 Raw data

Click here for additional data file.

Additional Information and Declarations

Competing Interests

Author Contributions

Patent Disclosures

Data Availability

The authors declare there are no competing interests.

Hang Pan conceived and designed the experiments, performed the experiments, analyzed the data, prepared figures and/or tables, authored or reviewed drafts of the article, and approved the final draft.

Yang Xiao conceived and designed the experiments, performed the experiments, analyzed the data, prepared figures and/or tables, authored or reviewed drafts of the article, and approved the final draft.

Ailin Xie conceived and designed the experiments, analyzed the data, prepared figures and/or tables, and approved the final draft.

Zhu Li conceived and designed the experiments, analyzed the data, authored or reviewed drafts of the article, and approved the final draft.

Haixia Ding conceived and designed the experiments, analyzed the data, authored or reviewed drafts of the article, and approved the final draft.

XiaoJu Yuan analyzed the data, authored or reviewed drafts of the article, and approved the final draft.

Ran Sun conceived and designed the experiments, analyzed the data, prepared figures and/or tables, and approved the final draft.

Qiuju Peng analyzed the data, prepared figures and/or tables, and approved the final draft.

The following patent dependencies were disclosed by the authors:

Bacillus megaterium L2 currently stored in the China Center for Type Culture Collection (CCTCC, NO. M2012381).

The following information was supplied regarding data availability:

The raw data are available in the Supplemental Files.

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
