# Peer review of "The antibacterial mechanism of phenylacetic acid isolated from Bacillus megaterium L2 against Agrobacterium tumefaciens"

_PeerJ, doi:10.7717/peerj.14304_

## Round 0.1 · original submission · Major Revisions

In addition to the reviewers' comments, the isolation and characterization of phenylacetic acid should be clearly highlighted.

·

Basic reporting

The manuscript is in general well-written with only a few minor points that need to be revised in the language used.
Please follow the structure of the article concerning affiliation, Department Name, Institution Name, City, State/Province, Country.
Also please delete numbring for subtitle and subheading.

In the abstract, please provide more details in Results part.
Figure 1 should be made with sufficient resolution.
In the introduction part, please provide more detail about Root cancer disease: Host plant, please name few host plants and the most important economically. For example, Conner and Dommisse.,1992 cited that A. tumefaciens can infest more that 93 families and 643 plant species.
Please provide more detail on How does Agrobacterium cause tumors?

In the introduction part, please provide sufficient information about Phenylacetic acid (PAA), for example, biosynthesis pathway of PAA, stucture as figure....

Experimental design

Methods used in this study provide sufficient information, and the investigation was conducted rigorously.
However, the statistical analysis must be developed with more details.
Please provide more detail about the statistical analysis for each experiment used in this study. One-way or multiple analysis of variance that are used? how many replication? The differences among groups were evaluated by performing Duncan’s test, please provide p-value of what is considered significant.

Validity of the findings

In the discussion part:
Its knows that the other compounds with antibacterial activity such as phenylacetic acid, behenic acid, and β-sitosterol can be considered as novel antibacterial agents. Please discuss the possibility of the Synergistic Activity of Bioactive compounds regarding increasing or decreasing their antibacterial activity.

In the conclusion part:
Please add informations regarding the industrial-scale of PAA, please add information about the availability of PAA, its advantage and limtation regarding fermentation yields and challenging accessibility via synthesis, also the possibility for development of genetically engineered strains and optimized cultivation processes.

Additional comments

'no comment'

·

Basic reporting

This paper is addressing an important topic regarding the antibacterial mechanism of Bacillus megaterium L2 phenylacetic acid against Agrobacterium tumefaciens.

Clear and unambiguous, professional English were used throughout, but grammar and punctuation should be checked.
Literature references, and concentrated field background was provided supported with recent references,
however the author should state a clear hypothesis, research questions and an intro about his work at the end of the introduction section.
This article shows a good Professional article structure including figures, tables.
The author self-contained with relevant results to hypotheses and it was clarified in the discussion section.

Experimental design

This is an original primary research within Aims and Scope of the journal.
Research question is well defined, relevant & meaningful. Yes, it stated how research fills an identified knowledge gap.

Investigations were performed to a high technical & ethical standard, however it was done on 2 strains only as a case study.

The author did not clarify the purification of PAA only from the extract? And did not clarify it in every section?, All the effect recorded can be due to other constituents in the extract other than the PAA.
The author used only 2 strains as a case study and generalized their facts, why no more samples were tested?
Why did the author did not perform in vivo tests, to validate their results?
Why was the bacteria growth measured at both O.D wavelength of 400 and 600 in several parts? and the bacterial usually measured at 600 nm and should be fixed in all methods.
Why was the chloramphenicol chosen as a control? it should be clarified.
How many ml of culture media were used? Ln98

Validity of the findings

Yes, meaningful replication encouraged where rationale & benefit to literature is clearly stated.

Yes, all underlying data have been provided; they are robust, statistically sound, & controlled.

Yes, conclusions are well stated, linked to original research question & limited to supporting results.

Additional comments

Title: Invitro analysis should be clarified.
Abstract:
The author should specify the methodology or select the most relevant, not to add etc
Introduction.
Discussion includes parts of detailed results with no clear discussion in several parts in page 12.
The punctuation and grammar should be checked.

Reviewer 3 ·

Basic reporting

The manuscript entitled " The antibacterial mechanism of Bacillus megaterium L2
phenylacetic acid against Agrobacterium tumefaciens" reported the evaluation of the known auxin and already petered out molecule (phenylacetic acid) against the causal agent of the crown gall of many dicotyledonous plants by studying involved mechanisms from three aspects: cell metabolism, biofilm integrity, and cell ultrastructure.

The manuscript is not well structured and prepared, and although it described several finding, is not convincing to be published in its form in this journal for the following reasons:

Abstract:
There is no clear strategy between the choice of involved mechanisms by phenylacetic acid for destroying the cells of the phytopathogen and its role as a means of biological control. If fact, there is a large gap between previous work and recent investigations of this molecule, and most research studies were focused on its role as a phytohormone rather than its use in biocontrol.

Introduction:
Lines 43-46: combines both sentences in one phrase to avoid redundancy.
Line 53: How you justify the use of fungicides and insecticides while the studied phytopathogenic agent is a bacterium?
Lines: 65-68
Bacillus megaterium L2 with a good biocontrol effect was isolated from the Asilixi Steppe in Bijie city in the early stage of the laboratory, its antibacterial activity against a variety of plant pathogens such as Ralstonia solanacearum and Alternaria alternata reached more than 90% (Yang et al., 2016; Ji et al., 2019)
Line 75: Erwinia carotii, I never heard about this phytopathogen? Among Erwinia, we found the following species, E. amylovora, E. chrysanthemi, E. carotovora…etc.
Material and methods:
Line 98: what do you mean by “an appropriate amount”?. You should define in order to get 108 CFU/mL after dilution and this constate is for all the text.
Lines 99-102: please clarify because there is ambiguity between the used concentration of PAA and chloramphenicol.
Line 110: write “the negative control” instead of “the blank treatment” for the whole text.
Line 114: you mean washing obtained bacterial cells obtained from culturing strain T-37.
Line 115: “bacterial culture” instead of “bacterial solution”
Lines 115-119: review the wording of the paragraph (PAA was added to….) which is not clear at all.
Lines 118 &126: What do you mean by “absorb the supernatant” please clarify how we can absorb the supernatant of culture filtrate after centrifugation
Lines 123-124 and line 125: the same comment as above, you should define the appropriate proportion.
Line 125: what do you mean by “bacterial liquid” do you mean “culture filtrate” or “culture fermentation”
Line 130-137: reformulate as fellow: The determination of the total sugar of the sample was conducted according to the method of Zhou et al. (2019). We draw first the standard curve using the concentration of glucose as the abscissa (mg/mL) and the absorbance as the ordinate. Then after, the bacterial suspension of T-37 was inoculated into NA medium at an appropriate…..
Line 138: with SpectroAmax TM 250 at 630 nm using sterile water as blank control….
Line 141: This analysis was performed using the method of Zhao et al. (2015), with some modifications:
Lines 145-149: reformulate the paragraph in order to be clearer.
Line 152: explain why you change the centrifugation parameter for the same culture: 4000 r/min for 10 min (line 145) then for 5 min?
Line 190: replaced by “Then the culture medium was inoculated with the bacterial suspension at concentration of ??? (that you should define)
Line 197: you should describe the characteristics of the machine

Results:
Line 205-213: write as fellow :
In this experiment, the inhibition rate of PAA on T-37 was determined at five concentrations of 0.6 mg/mL, 0.8 mg/mL, 1 mg/mL, 1.2 mg/mL and 1.4 mg/mL. Obtained results showed that at the inhibition rate of PAA reached 80.00 ± 0.02 % at the mass concentration of 1.0 mg/mL and no significant difference were observed at concentration of 1.2 mg/mL and 1.4 mg/mL (P <0.05). However, the inhibition rate of chloramphenicol reached 87.76 ± 0.01 % at the concentration of 0.2 mg/mL and was not significantly different from that at 0.4 mg/mL (P <0.05). Concerning the IC50 calculated by Graph Pad Prism 9 software was 0.8038 mg/mL and 0.0658 mg/mL respectively for the PAA and the chloramphenicol and those value were for subsequent experiments.
Line 221: “after 3 h of treatment” instead of “When treated at 3 h”
Lines: 226-227: leave the sentence for the discussion part: “indicating that PAA could affect the permeability of the T-37 cell membrane and release the cell contents into the culture medium for conductivity rises”.
Line 230: Explain how nucleic acid leakage from cells could accurate?
Lines 231, 239, 250, 260, 261: It is necessary to choose the duration after treatment with the PAA to compare with treatment using chloramphenicol and the control because you have shown that at 3h the electrical conductivity increases significantly then for the nucleic acid, the protein, and the sugar leakages you have found other intervals of time (4h and 2h). How you can explain these findings, otherwise, you should describe the chronology of occurred events.

Figure 6 : use the following legend for the SDS page patterns :

0h 4h 8h 12h
M C PAA Chlor C PAA Chlor C PAA Chlor C PAA Chlor

Line 285: It should be better that microscopic observations were done at different interval times to explain the significant differences between the treatment with PAA, chloramphenicol, and the control related to the other parameters

Overall: this paper describes several interesting results showing the role of treatment of T73 cells with PAA on ultrastructural, metabolic, and physiological changes in comparison with chloramphenicol and the control. However, it is not well prepared and, in its form, does not deserve to be published. The materials and methods section needs to provide more clear information. The discussion section needs to be improved and obtained results should be justified.
Moreover, the English writing of the paper needs to be revised.

Experimental design

The materials and methods section needs to provide more clear information.

Validity of the findings

Overall: this paper describes several interesting results showing the role of treatment of T73 cells with PAA on ultrastructural, metabolic, and physiological changes in comparison with chloramphenicol and the control. However, it is not well prepared and, in its form, does not deserve to be published.The discussion section needs to be improved and obtained results should be justified.

---

## Round 0.2 · Minor Revisions

In addition to the comments from the reviewers, please improve the English of the manuscript.

·

Basic reporting

The article is clear and well written in English and conform to professional standards of courtesy and expression. Literature references are sufficient and linked to field context of the reseach. Paper include sufficient introduction after revision. Good and Professional article structure.
However, I suggest to remove p-value from abstract.

Experimental design

Research question well defined; The investigation conducted rigorously.
Methods described with sufficient detail & information to replicate .

Validity of the findings

All data provided are robust, statistically.
Please note that conclusion should not contain references. Please remove the paragraph from 656 -677 to discussion part.
I suggest to combine the two parts (648 -654) and (678 - 683) in one paragraph as conclusion.

·

Basic reporting

The authors addressed the comments to improve the background.

Experimental design

The authors clarified the comments regarding each point and they gave explanation for choosing their experimental design
.

Validity of the findings

The manuscript's findings explain the mechanism of action of the PAA against strain T-37 based on scientific evidence and provide different perspective.

Reviewer 3 ·

Basic reporting

Dear Editor in chief
After reading the revised version and the response to the various raised comments, I conclude that the authors have clarified all comments and that the new version is well improved. It remains regarding the comment 4: Lines: 65-68 to modify "antibacterial" by "antimicrobial" as Alternaria alternata is not a bacterium
Comment 4: Lines: 65-68:Bacillus megaterium L2 with a good biocontrol effect was isolated from the Asilixi Steppe in Bijie city in the early stage of the laboratory, its antibacterial activity against a variety of plant pathogens such as Ralstonia solanacearum and Alternaria alternata reached more than 90% (Yang et al., 2016; Ji et al., 2019)
Response: Thanks for your nice comments. I'm very sorry, I don't quite understand how you want us to change this comment.
In conclusion, this article meets the PeerJ criteria and could be accepted.
I am at your total disposal for any other query regarding this manuscript
Cordially,
Mustapha Barakate

Experimental design

the material and methods section has been improved and the experimental protocols are well detailed in this new version

Validity of the findings

the interpretation and discussion of all obatnied results are now better written and more convincing

---

## Round 0.3 · accepted · Accept

Your manuscript is ready for publication.